# Investigation of Effects of Slider Structure on the Reversing Performance of Four-Way Reversing Valve

Kepeng Zhang [1], Dazhuan Wu [1,2,*], Huan Wang [1], Jianjun Li [3], Zhongbo Feng [3], Jiafeng Zhu [3] and Zhounian Lai [1]

1   Institute of Process Equipment, College of Energy Engineering, Zhejiang University, Hangzhou 310027, China; zhangkepeng@zju.edu.cn (K.Z.)
2   The State Key Laboratory of Fluid Power Transmission and Control, Zhejiang University, Hangzhou 310027, China
3   Zhejiang DunAn Artificial Environmental Equipment Co., Ltd., Zhuji 311835, China
*   Correspondence: wudazhuan@zju.edu.cn

**Abstract:** Unsmooth reversing is one of the most common faults in the four-way reversing valve of the air conditioning system, and the airflow in the valve chamber and the shape of the slider of the reversing valve are the main factors in unsmooth reversing. In order to study the influence of the airflow in the chamber and the slider shape on the reversing process, the fluid flow in the valve chamber of a four-way reversing valve is obtained using computational fluid dynamics (CFD) simulations, and the influence of the slider structure on slider thrust and resistance of the four-way reversing valve is analyzed. Four-way reversing valves with no-cutting, straight-cutting and arc-cutting sliders are employed to evaluate the reversing characteristics. A pressure measuring device for a four-way reversing valve chamber is designed, which is performed to test the chamber pressure of three different types of sliders of four-way reversing valves. The maximum error of experimental tests and simulations is within 5% of engineering tolerance, so as to verify the reliability of numerical simulation. The resistance-to-thrust ratio of the slider is raised to evaluate the performance of the reversing, which can guide the design of the valve and slider. The results indicate that the chamber pressure and resistance-thrust ratio with the cutting slider are reduced, which is beneficial to the reversing of the four-way reversing valve. Compared with the straight-cutting structure, the arc-cutting slider four-way reversing valve has a more stable reversing process and better comprehensive reversing performance. Results demonstrate that both the straight-cutting structure and arc-cutting structure of the slider can improve the process performance but the arc-cutting slider is better.

**Keywords:** four-way reversing valve; computational fluid dynamics (CFD); slider; reversing performance

## 1. Introduction

With the increasing importance of global greenhouse gas emissions and climate warming, the energy conservation and emission reduction of HVAC have been paid more and more attention by the relevant national departments and industries, which also put forward higher requirements for air conditioning accessories. The four-way reversing valves are the main component of the air conditioning system, which is mainly used in the heat pump type air conditioning unit. Through the four-way reversing valve, the air conditioning system can realize the conversion of refrigerant flow direction according to the needs, and change the functions of the condenser and evaporator of the air conditioning system, so as to realize the switch of the refrigeration, heating, defrosting and other functions of the air conditioner.

Researchers have done a lot of work on the characteristics of air conditioner four-way reversing valves [1–4]. Krishnan [5] evaluated the impact of the four-way reversing valves on the performance of the heat pump air conditioning system and pointed out that the poor performance of the four-way reversing valves could lead to a 5% reduction in the efficiency of the whole system. Damasceno [6] discussed the influence of the four-way

reversing valves on the heat pump air conditioning system through a mathematical model in terms of heat transfer, pressure drop and leakage. It was found that the pressure drop loss had the greatest impact on the system performance in the cooling condition, while in the heating condition, the pressure drop loss became the second impact factor. Raichintala and Kulkarni [7] also established a mathematical model of the four-way reversing valves to describe the energy loss of the valves and its influence on the heat pump system. In this mathematical model, energy loss caused by friction, refrigerant leakage and heat transfer was separated from the total energy loss and each factor was investigated separately. The results indicated that heating capacity, COP, and energy efficiency were related to the factor above all. The mathematical model isolated the pressure losses due to friction, pipe-fittings, mass-leakage and heat transfer from the total losses. The evaluation of constituent losses assisted in detecting a faculty reversing valve, and also determining the effect of mass leakage and heat leakage on the compressor work input and COP of the heat pump. Deng [8] studied and compared the performance of the heat pump system with and without four-way reversing valves by using experimental methods, and the results showed that the heating capacity, performance coefficient (COP) and energy efficiency of the heat pump system without four-way reversing valves were improved. Chen [9] analyzed the two basic conditions for smooth reversing of four-way reversing valves from theoretical and experimental perspectives. Liu [10] studied the influence of the flow channel on the flow performance of the four-way reversing valves through simulation and experiment. It was found that increasing the turning radius of the slider appropriately to reduce the resistance loss along the flow could achieve the purpose of improving the flow capacity of the four-way reversing valve. Peng [11] compared and analyzed the performance of the traditional four-way reversing valves structure and D-E straight-through four-way reversing valves, and the results showed that D-E straight-through four-way reversing valves could improve the heating performance of the system, but could not improve the annual energy consumption efficiency of the air conditioning system APF (Annual Performance Factor).

In the early stages, researchers conducted many studies on the impact of four-way reversing valves on energy efficiency, heat transfer and leakage of the whole air conditioning system, but rarely involved research on the reversing performance of four-way reversing valves [12,13]. As the slider is the main part of the four-way reversing valves, the research on the reversing performance of the slider to the four-way reversing valve is not comprehensive enough. There is no reasonable stable reversing scheme for reversing performance of four-way reversing valves.

In this study, the objective is to investigate the influence of different sliders on reversing performance during the reversing process. Furthermore, the ratio of resistance-to-thrust is raised to evaluate the reversing performance. This work intends to provide some references for the design of the slider in the four-way reversing valve.

## 2. The Problem Definition

The four-way reversing valves are located at the exhaust end of the air conditioning compressor. The four-way reversing valves realize the reversing (slider thrust) by pushing the slider in the valve body through the high temperature and high-pressure refrigerant gas from the compressor. At the same time, the slider is also under the pressure of the upper part of the high-pressure tube to achieve the fit between the valve seat (slider resistance). The ratio of slider resistance and slider thrust is defined as the resistance-to-thrust ratio of the slider. A small resistance-to-thrust ratio means that resistance is less than thrust, and reversing is mainly driven by thrust, thus the smaller the resistance-to-thrust ratio of the slider, the easier the reversing. On the contrary, a large resistance-to-thrust ratio means the resistance is larger than the thrust, and the resistance is caused by friction, which will bring vibration and noise problems to the valve body. Therefore, there has been more and more attention on the research of the structure of the slider of the four-way reversing valve and

the flow field in the valve body, especially the research on the pressure and velocity of the refrigerant gas near the slider in the reversing process.

The slider is one of the core components of the four-way reversing valves, which mainly has two functions: one is to switch the flow direction of the refrigerant when the four-way reversing valves are switched over in the cooling and heating cycle; the other is to isolate the high- and low-pressure areas in the non-reversing stable state. In the process of the slider moving from one end to the other end, there is a period of time between when the C tube, E tube and S tube are ventilated. When the slider is in the middle position, the flow between the C/E/S tubes is called the middle flow, which plays the role of pressure relief. The design of the intermediate flow is a critical factor that can determine the reversing performance of the four-way reversing valve. If the intermediate flow rate is too small, the refrigerant pressure in the valve body will rise instantaneously and impact the slider, the friction resistance of the slider will increase, and the vibration and noise of the valve body will be produced at the same time. Excessive intermediate flow will lead to a small pressure difference in the piston chamber, insufficient thrust, and reversing stagnation, affecting the normal reversing function of the four-way reversing valve. The structure of the slider directly affects the intermediate flow of the four-way reversing valves, so it is necessary to design a slider with appropriate thrust and friction resistance structure to ensure the reversing performance of the four-way reversing valves.

## 3. Structure and Principle of Four-Way Reversing Valve

### 3.1. Structure of Four-Way Reversing Valve

The four-way reversing valve is a complex component in the air conditioning system, which is mainly composed of the slider, valve seat, valve body, pilot valve, piston, slider support frame and tubes connecting the inlet and outlet of the compressor and the two units of the air conditioning system (evaporator and condenser), as shown in Figure 1. A slider embedded in the support frame changes the refrigerant flow direction from one end of the valve chamber to the other. The flow field region is considered for this study; besides the D/E/S/C tube, slider, and valve body, other parts are not going to appear in the simulation.

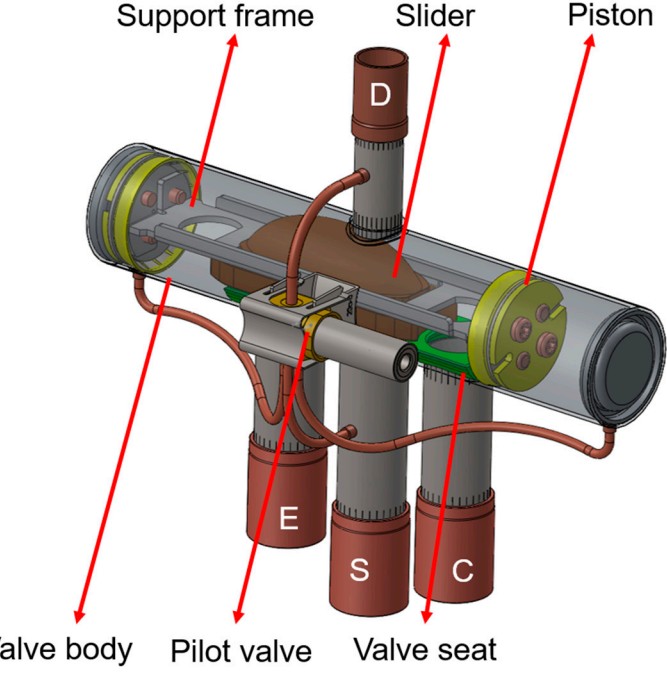

**Figure 1.** Three-dimensional model of four-way reversing valve.

### 3.2. Working Principle of Four Way Reversing Valve

The working principle of the four-way reversing valve is shown in Figure 2. When the heat pump air conditioning system switches from the heating cycle shown in Figure 2b to the cooling cycle shown in Figure 2a, the electromagnetic coil is powered off, the pilot slide valve moves to the left driven by the compression spring force, and the high-pressure refrigerant gas enters the right piston chamber through the pilot valve. At the other end, the compressor continuously draws gas into the compressor through the S-tube, reducing the pressure in the left piston chamber. At this time, there is a pressure difference between the piston chambers, thus moving the slider to the left. When the slider moves to the left end position, the E/S tube and the D/C tube are connected, respectively, and the heat pump system switches from the heating cycle to the cooling cycle.

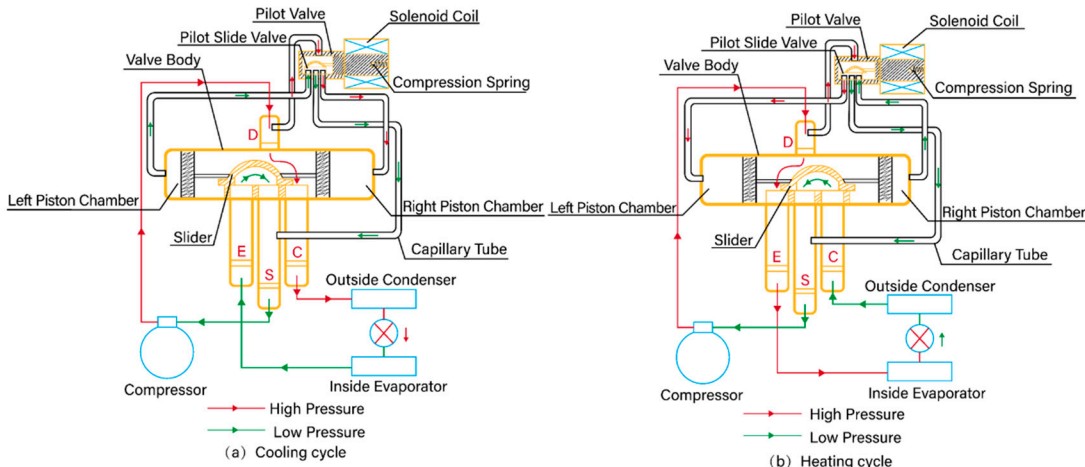

**Figure 2.** Schematic diagram of four-way reversing valve. (**a**) Cooling cycle; (**b**) Heating cycle.

When the electromagnetic coil is energized, the electromagnetic force generated by the electromagnetic coil in the pilot valve moves to the right to overcome the tension of the spring, and the high-pressure refrigerant gas enters the left piston chamber through the pilot valve. At the other end, the compressor constantly draws gas into the compressor through the S tube, reducing the pressure in the right piston chamber. At this time, there is a pressure difference between the left and right piston chambers, thus moving the slider to the right. When the slider moves to the right end position, the C/S tube and the D/E tube are connected, respectively, and the heat pump system switches from the cooling cycle to the heating cycle.

## 4. Hydrodynamic Control Equations

The two equations are applicable to the flow caused by the pressure difference. Due to the complex internal structure of the four-way reversing valve, the flow fields in the valve own turbulence properties with a high Reynolds number. In this paper, the Realizable *k-ε* turbulence model is employed for the simulation study [14–17].

The control equation of turbulence is three-dimensional incompressible Reynolds Averaged Navier Stokes (RANS) Equations. CFD simulations using the commercial software STAR-CCM+ 2022.1 are performed in order to obtain the pressure and velocity. The continuity, momentum and turbulence generation equations are solved by means of Finite Volume commercial code STAR-CCM+ [18].

## 5. Simulation Model and Boundary Conditions

### 5.1. Simulation Model

Since the structure and size of the slider are the main factors affecting the intermediate flow rate, the no-cutting, straight-cutting and arc-cutting slider structures are selected to analyze the reversing process; observe the changes in fluid pressure and speed in the valve

chamber, as well as the changes in the resistance and thrust received by the slider; and then analyze and compare the reversing performance of the four-way reversing valve. The specific structure is shown in Figure 3.

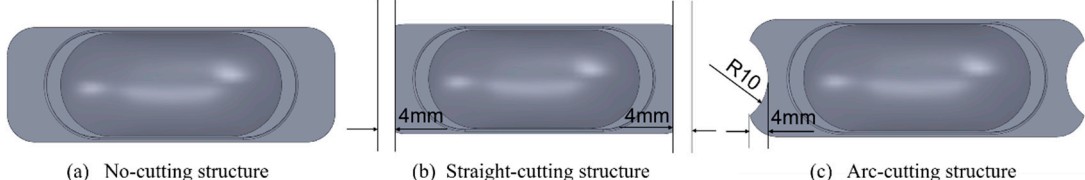

**Figure 3.** Three different slider structures.

The refrigerant flow area such as the valve body, slider, valve seat and C/E/S/D tube is taken as the calculation domain, while some non-essential parts such as the capillary and pilot valve are ignored. Figure 4 shows the grid cross-section diagram of the calculation model, which is a 3D model including all the fluid regions. The medium at the inlet (D tube) is from the compressor for the air conditioning system, which is the gas state with high temperature and pressure, while at the outlet (E-S tubes), the medium is a mixture of gas and liquid. The purpose of this article is the study of reversing performance, rather than the study of the two-phase flow; we have paid more attention to the pressure induced by the flow, thus, the medium is simplified to the gas state [19]. The calculated flow medium is refrigerant R410A, which is modeled in the gas state, and the corresponding thermal and physical parameters are obtained from the NIST REFPROP software, version 10.0. The turbulence model with Realizable $k$-$\varepsilon$ Reynolds Averaged Navier Stokes (RANS) Equations is addressed; it has the characteristics of high precision and low consumption of computing resources. The Enhanced Wall treatment is employed for the simulation, which combines the two-layer near-wall model with enhanced wall functions. If the near-wall mesh is fine enough to be able to resolve the laminar sublayer, the Enhanced Wall treatment solves the whole turbulence boundary layer, including the viscous sublayer. The Enhanced Wall treatment allows for a near-wall formulation that can be used with coarse meshes as well as fine meshes, which has good mesh adaptability. The main advantage of polyhedral mesh is that it has multiple adjacent cells to reduce the total number of grids, and the calculation accuracy can be well guaranteed. In order to simulate the sliding process of the slider, the sliding grid technique is adopted. In the sliding grid technique, two or more cell zones are used, and each cell zone is bounded by at least one "interface zone "where it meets the opposing cell zone. The interface zones of adjacent cell zones are associated with one another to form a "mesh interface" The two cell zones will move relative to each other along the mesh interface. During the calculation, the cell zones slide relative to one another along the mesh interface in discrete steps.

*5.2. Grid Independency Analysis*

The number and quality of meshes is the critical factor to determine whether the results are reasonable or not. In order to ensure the accuracy of the grid, it is necessary to check the independency of the grid. Considering the main purpose of this study, the no-cutting structure and mass flow rate are chosen as the parameters to check the grid independency. The grid independence test results are shown in Figure 5. It can be seen that when the number of grids is 5,200,000, the flow rate error is within 0.2%; when the number of grids is more than 5,200,000, the error continues to decrease, but the calculation time continues to increase. In consideration of accuracy and calculation time, 5,200,000 grids are selected for this study.

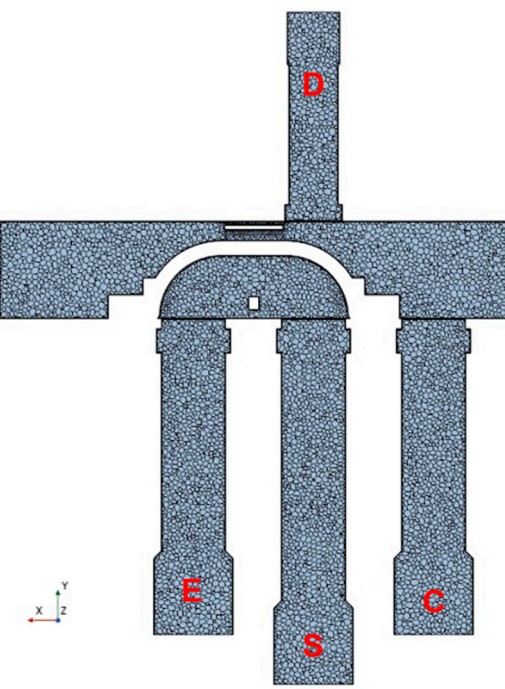

**Figure 4.** Mesh of computational model.

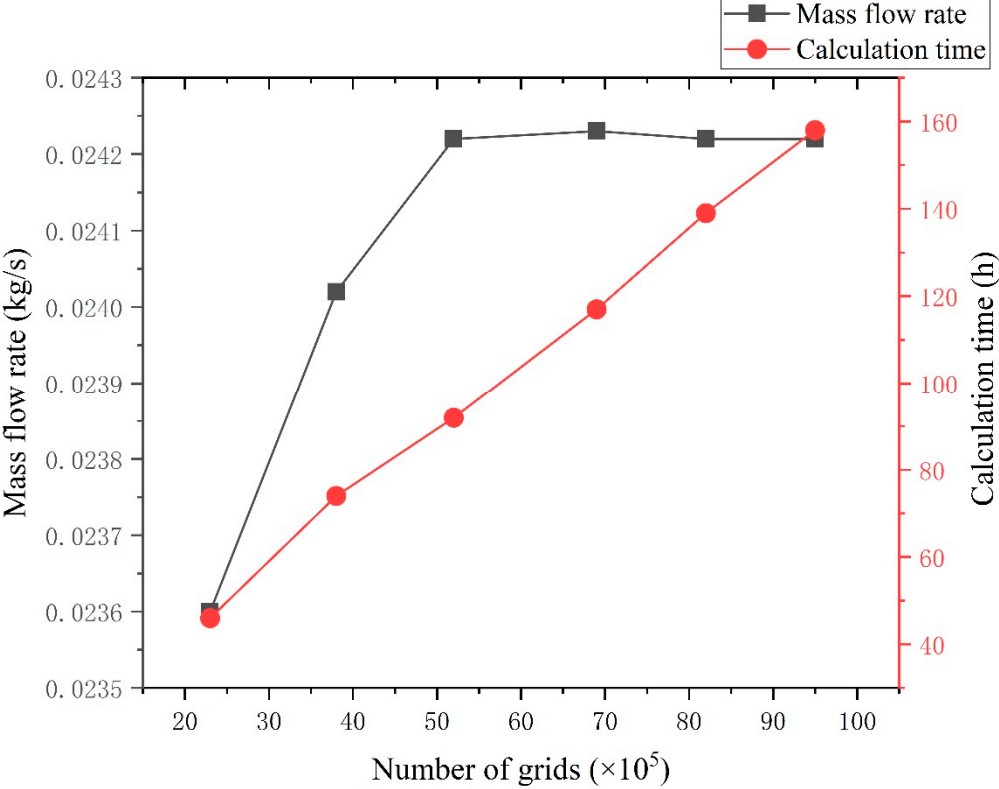

**Figure 5.** Grid independence analysis.

### 5.3. Time Step Sensitivity Analysis

In transient calculation, if the time step is too large, the calculation will diverge; on the contrary, the calculation time will increase when the time step is too small. Therefore, it is necessary to analyze the sensitivity of the time step. The time step sensitivity analysis results are shown in Figure 6. It can be seen that the mass flow keeps fewer changes with

the time step decreasing. When the time step is $1 \times 10^{-6}$ s, the mass flow is considered to be stable. If the time step is too small, the calculation time is greatly increased. As a consequence, the time step of $1 \times 10^{-6}$ s is selected for the balance of numerical accuracy and efficiency.

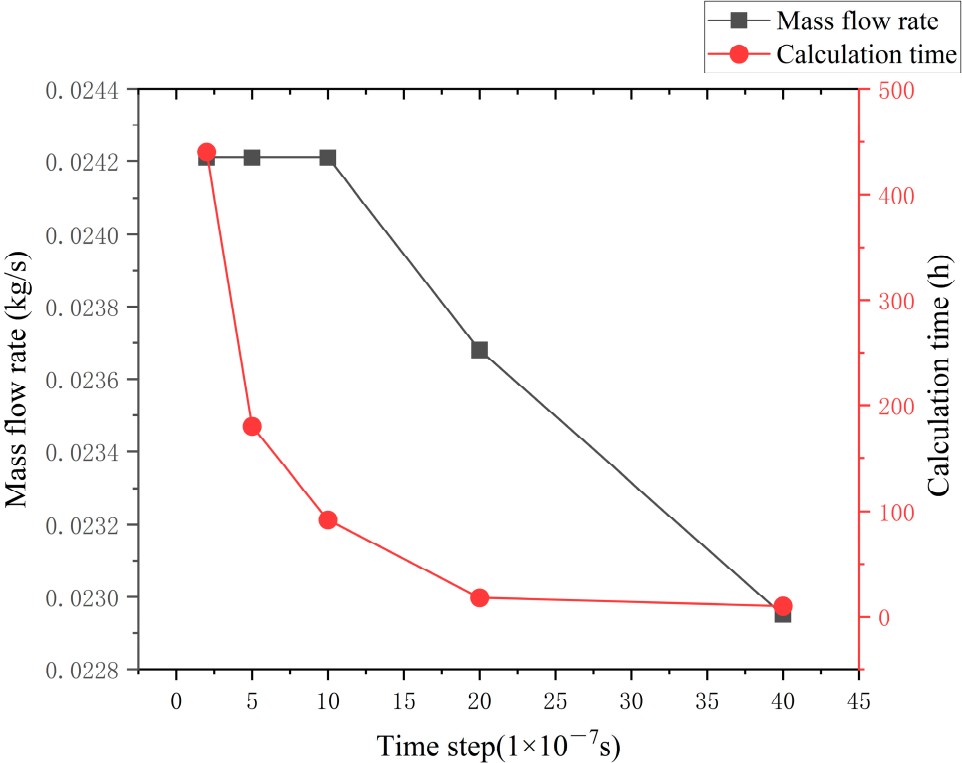

**Figure 6.** Time step sensitivity analysis.

### 5.4. Boundary Conditions

In order to ensure the consistency between the numerical simulation and the actual system, the geometric model, boundary conditions and reversing time are measured from the actual product operation. The E tube and C tube are the pressure outlet boundary. The pressure of the C tube is slowly reduced from 2.9 Mpa to 0.5 Mpa by using the dynamic function, which is the opposite of E tube. The D inlet tube is set at the stagnation inlet boundary of 3.0 Mpa. The outlet tube S is set as the outlet boundary condition of 0.4 Mpa, and the other wall boundary is set as the wall boundary, as shown in Figure 7. The values of boundary conditions are based on the real AC system, and the pressure is relative pressure, namely gauge pressure. In order to ensure rapid convergence and calculation speed, the time step is set to $1 \times 10^{-6}$ s. The simulation process is from the cooling cycle to heating cycle, while the slider slides are from the E tube to C tube. The dynamic pressure functions are loaded using TABLE in STAR-CCM+, and the dynamic pressure function of the C tube is presented as follows:

$$P_t = -6t + 2.9 \tag{1}$$

where $P_t$ is the outlet pressure of C tube, Mpa; $t$ is the reversing time, s.

### 5.5. Experimental Comparison and Analysis

In order to verify the authenticity of the simulation, a pressure test bench for the valve chamber of the four-way reversing valve was designed and built. In order to facilitate the arrangement of sensors, a pressure sensor was placed on the chamber wall above the E tube in the valve chamber of the four-way reversing valves, which is used to measure the change of pressure at this point during the reversing process of the four-way reversing valve. The D tube connects to the outlet of the compressor, and the E tube and C tube

connect to the inlet of the evaporator and condenser, respectively, while the S Tube connects to the inlet of the compressor. Besides the valve itself, the refrigerant impurities may be an uncertain factor in the test. Filters are added to the air conditioning system to filter impurities and ensure the accuracy of the test. The experimental platform is horizontal, as shown in Figure 8.

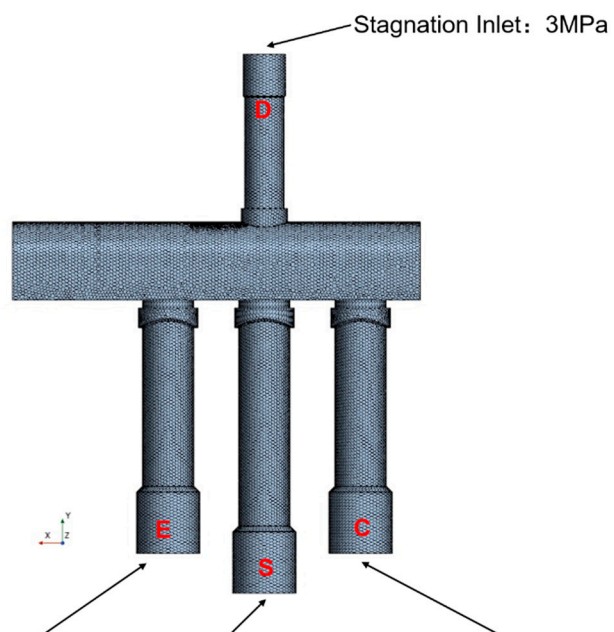

**Figure 7.** Computational boundary condition.

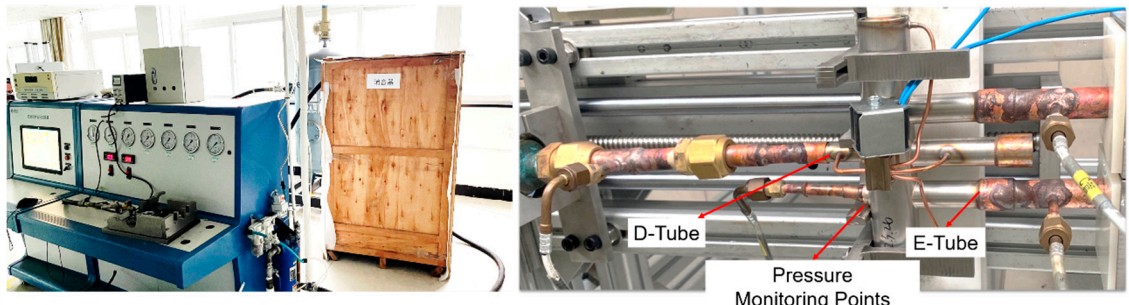

**Figure 8.** Acquisition experimental platform of pressure.

After the reversing stability of the four-way reversing valves, the experiment was carried out to eliminate the unstable factors at the beginning and near the end of the reversing. The count was from 0.1 s of each reversing to 0.3 s before the end. The pressure value was taken every 0.05 s, the pressure was measured 5 times and the average value was taken. The pressure results at this point were extracted from the pressure field results of three slider structures, respectively, and compared with the average test results of three different slider structures. The results are shown as Figure 9, and the values on the lines are the difference between experiment and simulation.

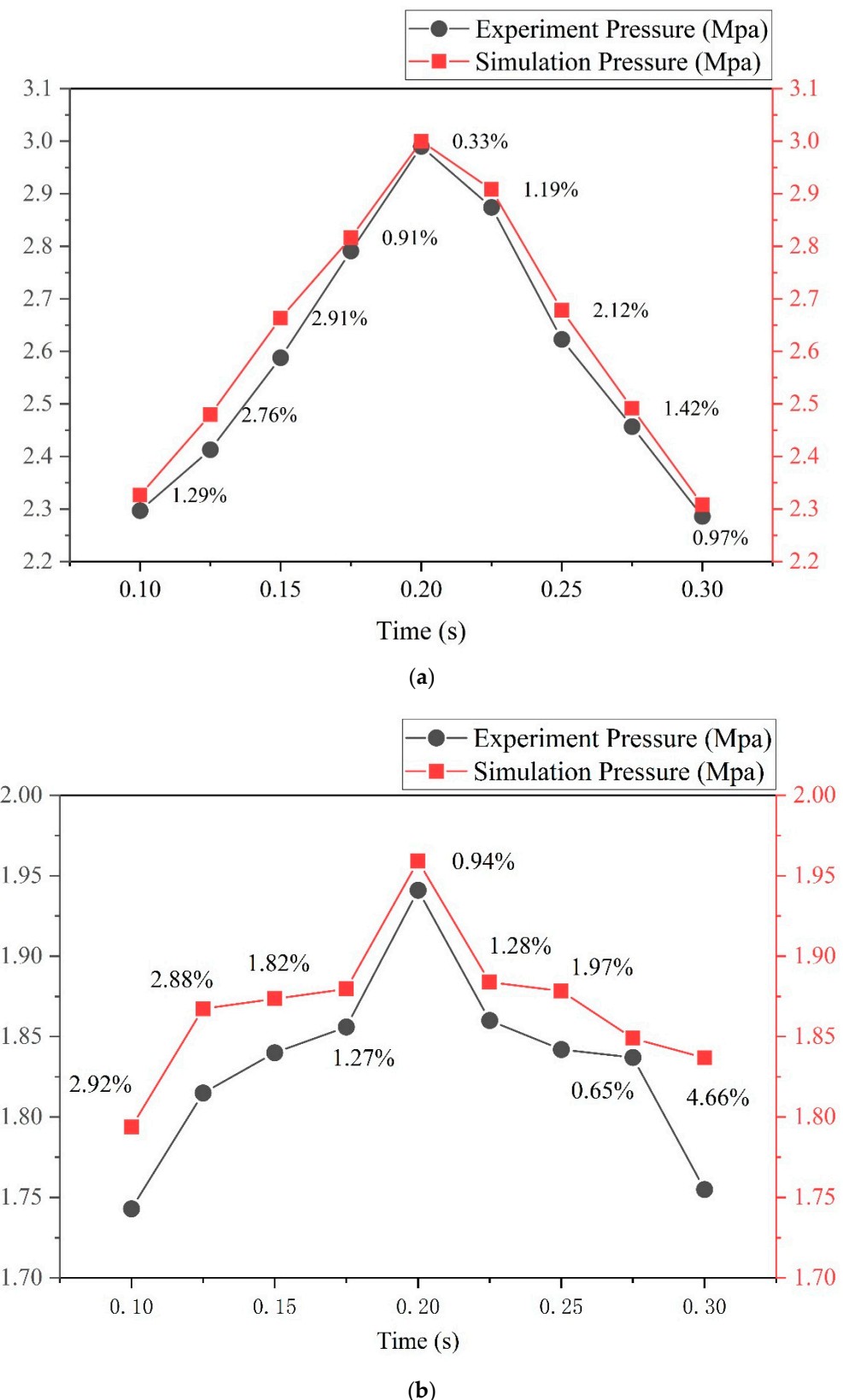

(**a**)

(**b**)

**Figure 9.** *Cont.*

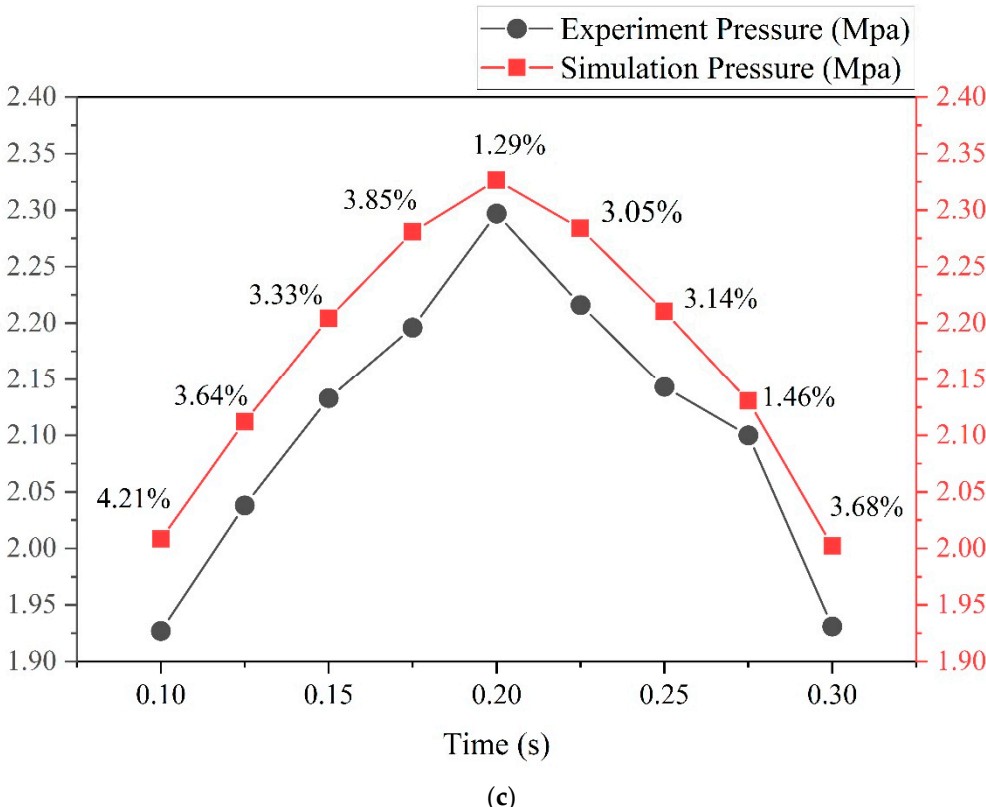

**Figure 9.** Results of experiment and numerical simulation on the pressure. (**a**) No-cutting structure; (**b**) Straight-cutting structure; (**c**) Arc-cutting structure.

As can be seen from the figure, the pressure variation trend calculated by numerical simulation tends to be consistent with the test results. The maximum error is 4.66% when the slider moves 0.3 s in the straight-cutting structure, and the allowable error range of engineering is ±5%. Therefore, the results obtained by numerical simulation are reliable and meet the accuracy requirements.

## 6. Results and Discussion

### 6.1. Analysis of Simulation Results

The slider structure determines the intermediate flow of the four-way reversing valves, and then affects its reversing fluent performance. The intermediate flow also affects the pressure and velocity of the refrigerant fluid in the valve body.

In the case of three different slider structures, the distribution of flow field pressure and velocity of refrigerant fluid in the four-way reversing valves chamber are shown in Figures 10 and 11 when the slider is under the initial sliding state, and the "start position" is in cooling mode.

As can be seen from Figure 10, the pressure value, distribution of high- and low-pressure areas, and pressure drop areas in the valve chamber of the different sliders are basically the same. After the high-pressure fluid is discharged from the compressor, it enters the valve body from the D tube, and then flows out to the two devices (condenser and evaporator) through the C tube, enters the E tube, and then enters the compressor to the next cycle through the S tube. The pressure distribution trend of the three structures also conforms to the above circulation process. The fluid pressure in the D tube is higher, and after entering the valve chamber, the fluid flows to both ends of the slider. On the left side of the chamber, the pressure is higher because the space is closed. The E tube and S tube are connected, belonging to the low-pressure side. Fluid flows from the E tube to S tube with the direction of flow changing 180° near the slider, and there are vortices near the direction change of flow, which can cause a pressure loss, resulting in a pressure drop. For

the three slider structures, the overall pressure distribution in the four-way reversing valves is as follows: The D tube has the highest pressure, due to the flow from the compressor directly. After entering the main chamber, the pressure far away from the port of the two devices (evaporator and condenser) is higher, while the pressure near the port of the two devices is lower. There are high-pressure and low-pressure regions in the S tube. The refrigerant flows from the E tube to S tube because of the pressure difference, and for the S tube there are nearly half high-pressure regions and half low-pressure regions. When the high-speed fluid flows through the inside of the slider, the flow direction of the fluid near the E tube changes by 180° dramatically, forming high-speed flow vortexes, accompanied by energy loss, which results in pressure reduction and the low-pressure region being formed. Meanwhile, the fluid far from the E tube whose flow direction changes by 180° gradually and the high-pressure region is formed. There is no significant difference in the pressure on the external surface of inside of the slider.

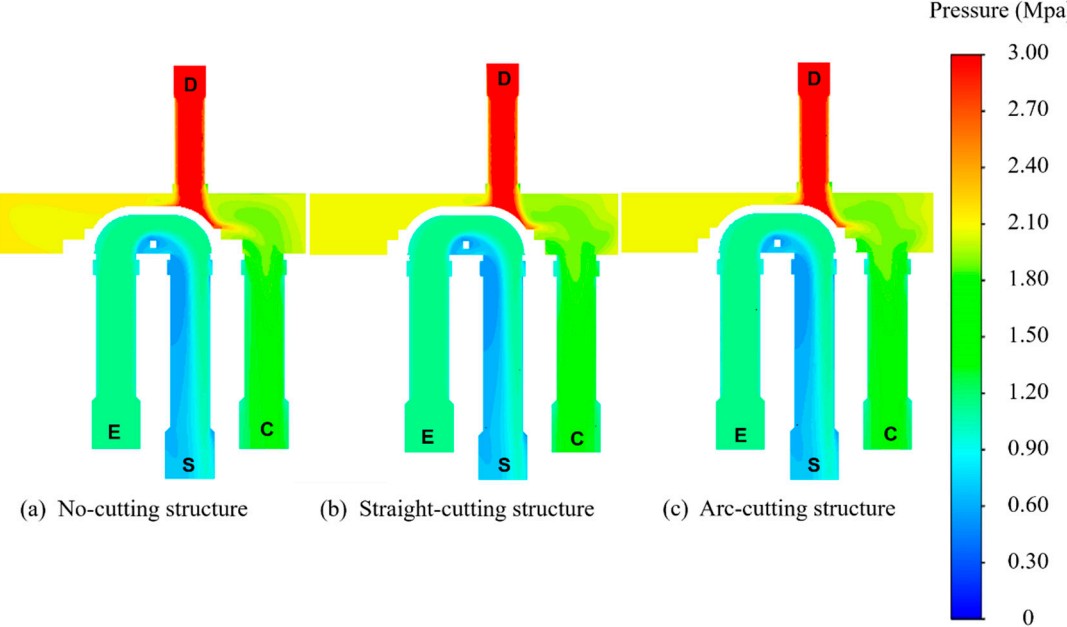

**Figure 10.** Pressure field of start position.

As can be seen from Figure 10, velocity distribution in the valve body cavity is similar to pressure for sliders of different structures, and velocity gradient changes in high-speed areas are basically consistent. The overall velocity distribution is as follows: After entering the main chamber, the velocity of the D tube is the highest. As there is no flow outlet near the two connecting tubes, the velocity in some areas is close to 0, and there is a flow dead zone, the fluid velocity in the whole valve body is low. Near the connecting pipe of the two tubes, the fluid velocity has a certain increase, and the discharge effect is obvious. Near the wall surface close to the S tube, due to the diversion and the change of flow direction, the fluid velocity is greatly improved, and there is no significant difference in the velocity on the external surface inside of the slider. The reason for velocity distribution is the same as that for pressure distribution. It is due to the 180° change of flow direction, and flow stagnant wake is generated due to the abrupt change of flow direction.

Since the intermediate flow rate is the flow rate when the slider slides to the middle position, the fluid condition in the cavity is the main factor reflecting the reversing performance. Therefore, it is necessary to investigate the distribution of fluid pressure and velocity in the cavity when the slider is in the middle position. Figures 12 and 13 show the distribution of flow field pressure and velocity in the chamber when the slider is in the middle position.

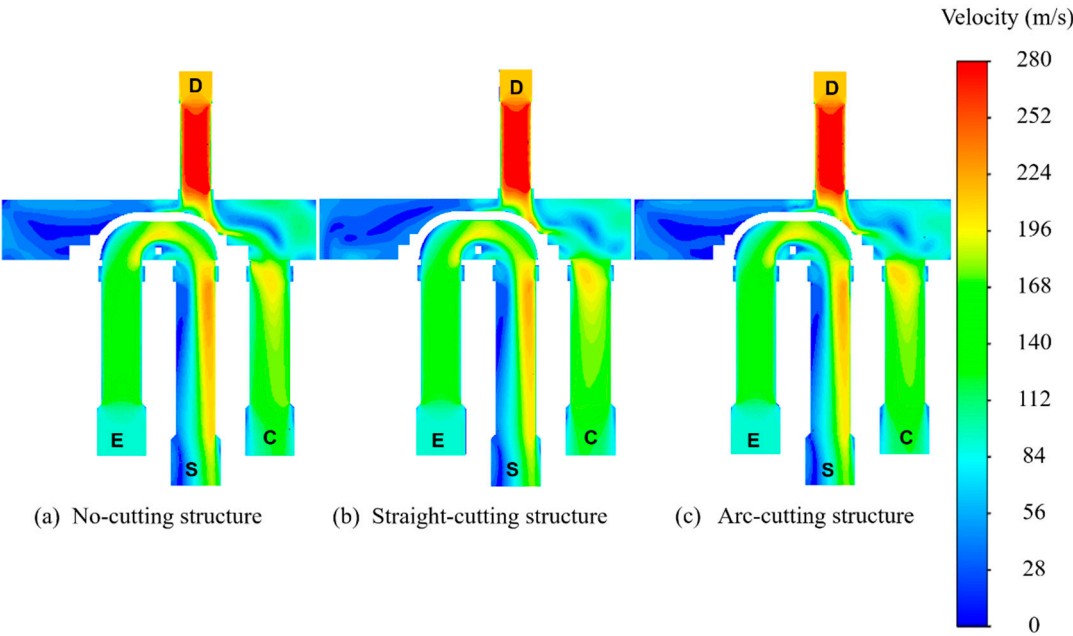

**Figure 11.** Velocity contours of start position.

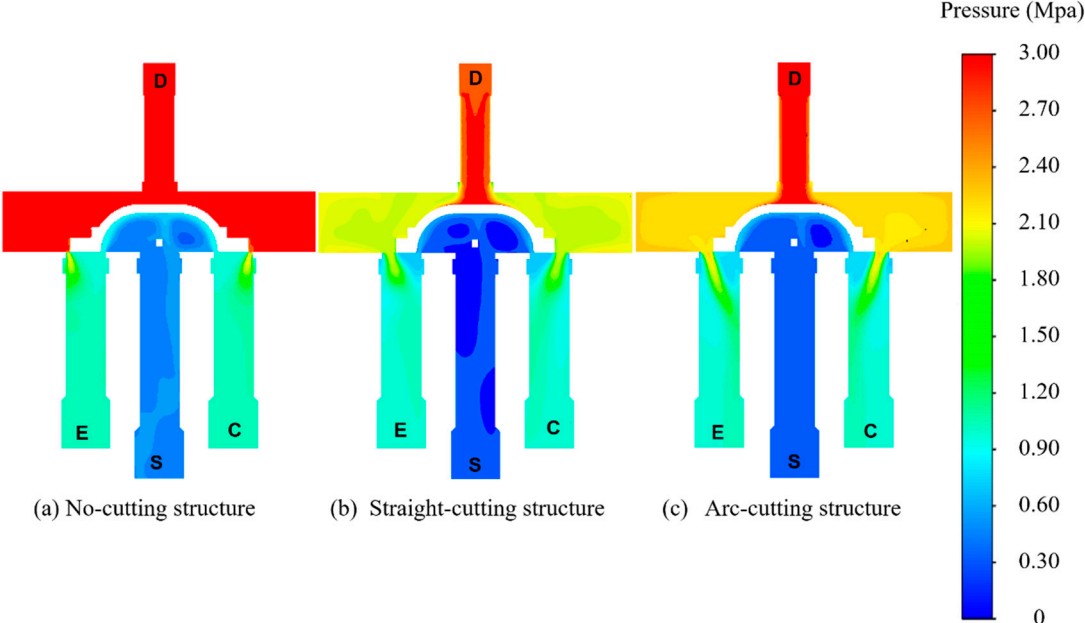

**Figure 12.** Pressure contours of middle position.

It can be seen from Figure 12, when the slider is in the middle position, the pressure in the chamber of the slider no-cutting structure is the largest, and the pressure at each position in the chamber is almost equal to that of the D tube, at about 3 Mpa. The pressure in the chamber of the slider with straight-cutting and arc-cutting structures is relatively small, at about 2.2 Mpa, which is obviously lower than that of the four-way reversing valve no-cutting structure. The pressure distribution in the whole chamber is relatively uniform. In addition, it can be seen from the pressure field that on both sides of the slider, the high-pressure release area of the straight-cutting structure and the arc-cutting structure is obviously larger than that of the no-cutting structure, which is also the main reason why the pressure distribution in the chamber of the cutting structure is lower than that of the no-cutting structure.

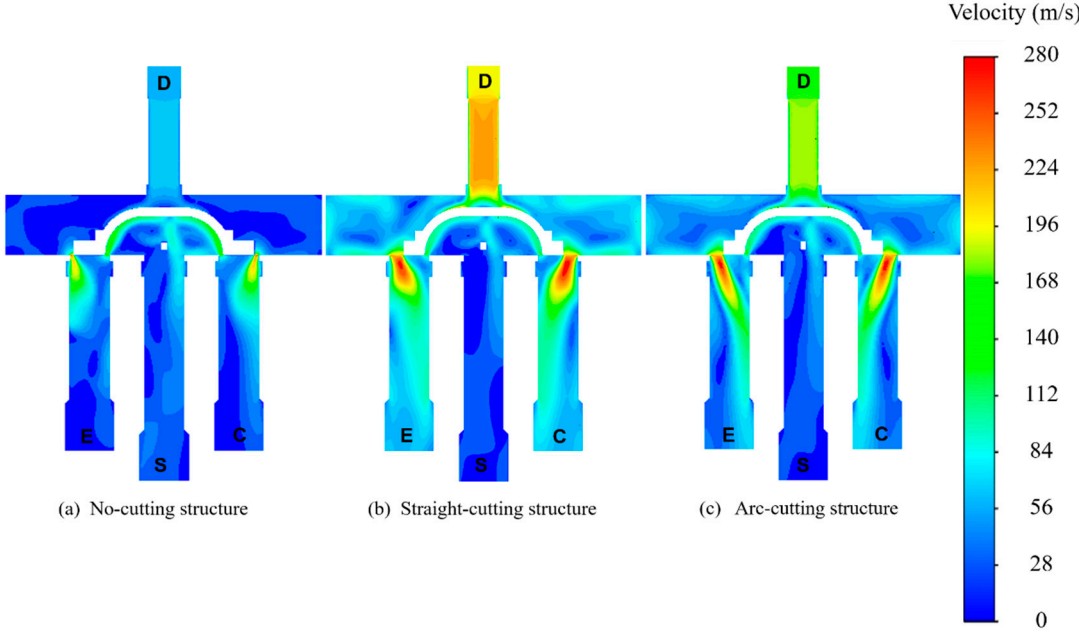

**Figure 13.** Velocity contours of middle position.

It can also be seen from the distribution of flow field velocity in Figure 13 that the fluid velocity in the chamber of the slider with cutting-structure is higher than that of the no-cutting structure. The slider with cutting-structure on both sides of the slider has an obvious velocity gradient, corresponding to the pressure cloud diagram. For the no-cutting structure, the pressure in the chamber is lower on the whole, and there is a certain flow dead zone, which is not conducive to fluid flow.

The results reveal that the process performance of the cutting structure of the slider is better than the no-cutting structure. However, it is not possible to judge which cutting structure has better performance from velocity and pressure distribution. Due to the high-pressure in the middle position and the resulting high friction resistance, the no-cutting structure may lead to poor reversing performance, while the low-pressure in the middle position and the resulting low thrust, the cutting structure could cause the slider to get stuck in the middle position.

### 6.2. Further Investigation of Slider Structure on the Reversing Performance

The basic condition of the four-way reversing valves is that the thrust $F_t$ of the slider is greater than the friction between the slider and the valve seat $f$, $f = \mu F_r$. $F_r$ is the positive pressure of the slider, $\mu$ is the kinetic friction factor, only related to the material and roughness of the contact surface, $F_r$ is defined here as the resistance of the slider. Therefore, the reversing performance of the reversing valves depends on the thrust $F_t$ and resistance $F_r$ of the slider [19].

$$F_t = (P_l - P_r)S_C \tag{2}$$

where $P_l$ and $P_r$ are the pressure on the left and right ends of the slider, Mpa; $S_c$ is the axial cross-sectional area of the slider, m$^2$.

$$F_r = P_t S_t - P_b S_b \tag{3}$$

where $P_t$ and $P_b$ are the pressure on the upper and lower surfaces of the slider, Mpa; $S_t$ and $S_b$ are the areas of the upper and lower surfaces of the slider, m$^2$.

In order to further study the reversing performance for the four-way reversing valve with different slider structures and find out which cutting structure has better performance, it is necessary to research the resistance and thrust acting on the slider of the valve.

The resistance and thrust changes of three slider structures in the process of reversing, that are from simulation results, are shown in Figure 14. It is obvious from the figure that the thrust of the three sliders is almost equal at the beginning and end of reversing. With the progress of the reversing process, the thrust of the no-cutting sliders increases until at the middle position where the value of the thrust gets the maximum 2316.6 N, and then decreases until the end of reversing. The thrust of the straight-cutting slider increases at the start position, getting the maximum value 1573.7 N at 0.12 s and then it decreases until at the middle position, and the trend from middle position to end is symmetrical to the front part. The thrust of the arc-cutting slider increases at the beginning until at 0.12 s position, where the value stays the same, which is 1766.5 N, and decreases from 0.28 s until the end of reversing.

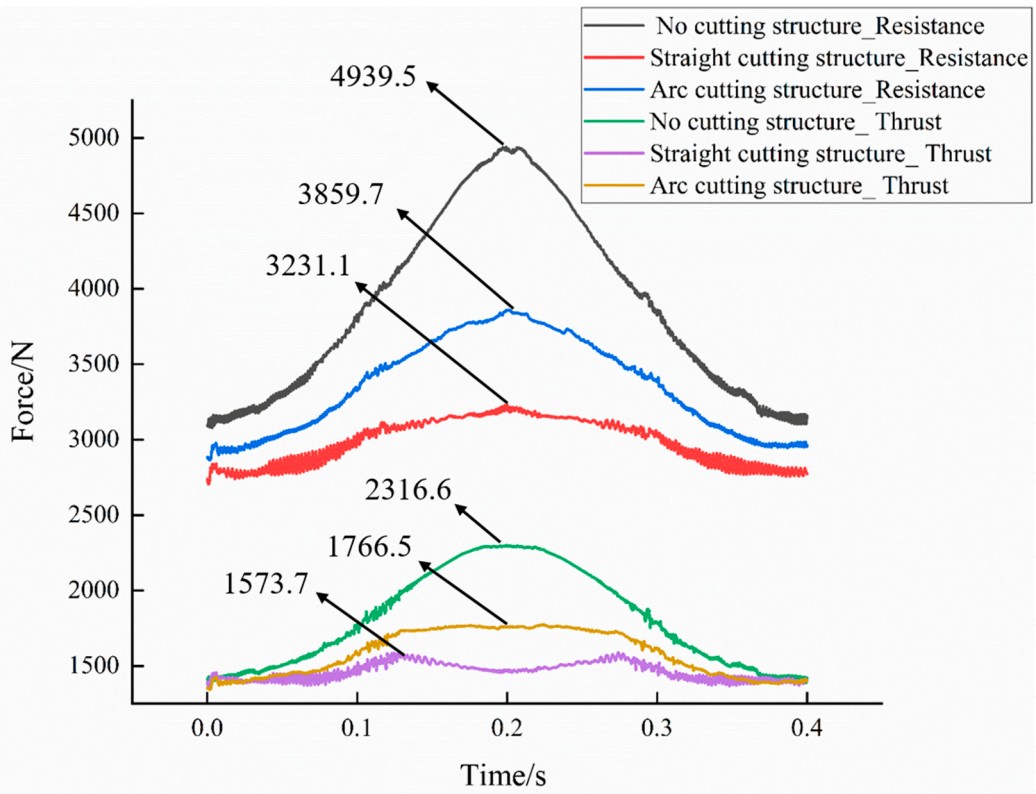

**Figure 14.** Resistance and thrust on slider.

It can also be seen from Figures 8 and 10, the pressure in the chamber of the three slider structures is similar at the beginning. When the slider reaches the middle position, the pressure in the chamber of the no-cutting structure is the largest, while the pressure in the chamber of the straight-cutting and arc-cutting structure is lower. It can also be seen from Figure 14 that the resistance of the no-cutting structure is the largest, followed by the arc-cutting structure and the straight-cutting structure. In addition, the trend of resistance in the process of reversing increases until the middle position with the maximum value and decreases until the end of reversing. This is because as the slider moves, the surface of the slider impacted directly by the high-pressure fluid is getting larger until the slider moves at the middle position with the surface reaching the maximum. After that, the surface of slider impacted by the high-pressure gradually decreases until the end of the reversing. The resistance of the non-cutting structure slider is the largest, because the surface of the slider impacted by the high-pressure for the non-cutting structure slider is the largest.

Since the resistance and thrust curves of the three structural sliders in Figure 14 are consistent, the reversing performance cannot be directly judged. In order to further compare the reversing performance of the three structures, the reversing process of the four-way reversing valve is observed by using the ratio of slider resistance and thrust (resistance-

to-thrust ratio). The structure which has better reversing performance is the smaller resistance-to-thrust ratio. The ratio of resistance-to-thrust of the three slider structures is shown in Figure 15.

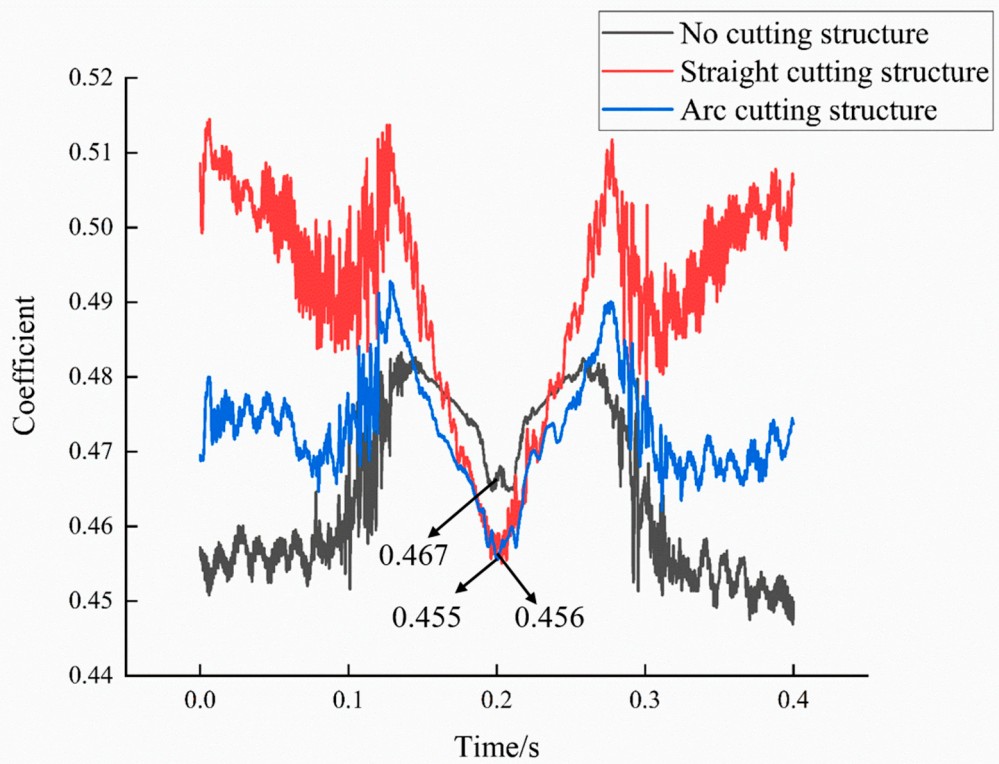

**Figure 15.** Ratio of the resistance-to-thrust on sliders.

It is obvious from the figure that the resistance-to-thrust ratio of the slider of the no-cutting structure is the smallest at the beginning and end of the reversing stage, followed by the arc-cutting structure; and the resistance-to-thrust ratio of the straight-cutting structure is the largest. However, at the middle position, the ratio of the resistance-to-thrust of the no-cutting structure is 0.467, which is the largest. The values of the resistance-to-thrust ratio of the arc-cutting structure and the straight-cutting are 0.455 and 0.456, respectively, and both are smaller than the no-cutting structure. As the middle position is the most important position in the whole reversing process, the pressure on the surface of the slider is the largest at this time and the reversing is the most difficult. Therefore, the reversing performance of the cutting structure slider is better than the no-cutting structure slider. In addition, it can be seen from the figure that the ratio of the resistance-to-thrust curve of the straight-cutting structure fluctuates more violently than the arc-cutting structure, which easily generates vibration and noise in the reversing process; in addition, the ratio of the resistance-to-thrust for the arc-cutting structure at the middle positon is smaller than the straight-cutting structure. Therefore, the reversing performance of the arc-cutting slider is optimal.

## 7. Conclusions

In this paper, the internal flow field of four-way reversing valves with different sliders and its influence on the reversing performance are studied by means of numerical analysis. It is found that the slider structure of the four-way reversing valves can cause changes in the flow field inside the chamber, and by comparing the resistance and thrust of different slider structures, it is found that the cutting structure slider valves are effective in improving reversing smoothness. The research results of this paper can provide a reference for the optimized design of the four-way reversing valves. It can be of important significance to improve the smoothness of reversing during the operation of air conditioners.

(1) The slider structure affects the pressure and velocity distribution in the valve chamber during the reversing process of the four-way reversing valve. At the beginning of reversing, there is little difference in the pressure and velocity distribution in the valve chamber of the no-cutting slider, straight-cutting and arc-cutting slider. When the slider slides to the middle position of the four-way reversing valve, the pressure in the chamber of the no-cutting structure slider is greater than that in the chamber of the cutting structure slider;

(2) At the beginning and end of the reversing, the thrust of the three structures is almost the same. The thrust of the no-cutting sliders increases until at the middle position where the value of the thrust gets the maximum 2316.6 N, and then decreases until the end of reversing. The thrust of the straight-cutting slider increases at the start position, getting the maximum value 1573.7 N at 0.12 s and then it decreases until at the middle position, and then increases until at 0.28 s, after which it decreases until the end. The thrust of the arc-cutting slider increases at the beginning until at 0.12 s position, and gets a maximum value 1766.5 N from 0.12 s to 0.28 s.

In the middle position of the reversing process, the ratio of the resistance-to-thrust for the no-cutting structure is the largest, with a value of 0.467. The ratio of the resistance-to-thrust for the arc-cutting structure is 0.455, which is the smallest among the three structures; thus, the reversing performance of the arc-cutting slider is the best scheme.

**Author Contributions:** Conceptualization, K.Z. and D.W.; methodology, K.Z., D.W., H.W. and Z.L.; software, K.Z. and H.W.; validation, K.Z., J.Z. and H.W.; formal analysis, H.W.; investigation, K.Z. and Z.L.; resources, J.L. and Z.F.; data curation, J.Z.; writing—original draft preparation, K.Z.; writing—review and editing, K.Z; visualization, K.Z. and H.W.; supervision, D.W.; project administration, Z.F. and J.L.; funding acquisition, D.W. All authors have read and agreed to the published version of the manuscript.

**Funding:** This research was funded by the National Natural Science Foundation of China (51839010), and The APC was funded by the National Natural Science Foundation of China (51839010).

**Data Availability Statement:** The data used to support the findings of this study are included within the article.

**Conflicts of Interest:** The authors declare no conflict of interest.

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
