# Peer review of "Investigation of Effects of Slider Structure on the Reversing Performance of Four-Way Reversing Valve"

_processes, doi:10.3390/pr11051538_

Round 1
Reviewer 1 Report
In this paper, the authors present numerical simulations and experiment for the fluid flow in the chamber of a four-way reversing valve. I suggest a minor revision and have the following comments.
1. The last sentence in the abstract is not suitable.
2. More information on computations needs to be provided. How many meshes are in the simulations? How about the CPU time for each case? Did the authors do the grid independence study?
3. For the fluid flow, streamlines can be inserted.
4. In Figure 11, it's better to plot two Y axis, to add a clearer illustration of the difference between experiment and simulation.
5. The border of each figure is not unifying and all the figures can be improved.
There are some grammar mistakes and the English can be improved.
Author Response
Please see the attachment。

Reviewer 2 Report
1. The English of the paper must be checked and improved. Poor English and a lack of technical academic language prevail in the paper. Some parts of the paper are unclear due to poor English.
Examples are:
· Section 3, the second paragraph is unclear mainly due to poor English. There is also no need for the first paragraph. The authors only need to present the governing equations correctly.
· In section 4, the statement “The easier the reversing, the resistance to push ratio is too large, which will cause poor reversing…” is unclear.
· In section 4.1, the statement “Whether the 165 design of intermediate flow is reasonable or not directly determines the reversing performance of four-way reversing valves” is unclear.
· In the paragraph below Figure 7, it is mentioned that “Near the wall surface close to the S-tube, the fluid velocity is greatly improved”. What do you mean by improvement since there are no bases for improvement?
· Above Figure 4, the governing equations are widely known as the Reynolds Averaged Navier Stokes (RANS) Equations and the term “Reynolds mean stress Navier-Stokes equations”, mentioned by the authors, is not a common terminology.
· Above Figure 12, “This is because with the movement of the slider, the slider surface is directly affected by the impact area of high pressure fluid more and more large, reached the middle position when the maximum, and then gradually reduced until the end of the reversing, and the effective compression area of the no-cutting structure is the largest, so the resistance of the no-cutting structure slider is the largest.” The sentence is too long and unclear.
2. The intention of conducting this study is unclear. The authors meant to evaluate the performance of a reversing valve with a different design; however, they did not mention what parameters to study and evaluate. They also did not state how the study contribute to the advancement of the field and how the results can help with the improvement of the valve. What are the pros and cons of each design and which one has the better performance?
3. The introduction section needs to address the significant findings of the previous studies, not only mentioning the type of study. For instance,
· mention the significant results of Damasceno [6],
· the percentage change of heating capacity, COP, and energy efficiency in Raichintala and Kulkarni [7],
· significant findings Raichintala and Kulkarni [7],
· noting was mentioned about the findings of Liu [10]
4. In the introduction section paragraph 3, the statement “… the reversing performance of the slide block to the four-way reversing valve is not comprehensive enough.” Need more explanation. What is the intention of authors by mentioning “it is not comprehensive enough.”
5. The last paragraph of the introduction section must be in the conclusion. This paragraph is supposed to address the novelty of the study which cannot be seen in the paper.
6. In Figure 1, what is the function of the other parts of the valve? Are they going to appear in the simulation? Mention the importance of each part of the valve concerning the study.
7. The author used the Realizable k-ε turbulence model. This needs to be cited.
8. How was the near wall region treated in the turbulent mode?
9. How were Reynolds stresses (mentioned as viscus stress by the author) modelled? Provide mathematical relations, please.
10. What do you mean by “the physical strength of the micro element”? how was it modelled? From the results and simulations, I do not see any microelement. The forces in the momentum equations are external forces which do not exist in your problem.
11. Was the internal heat source of fluid modelled? I do not see any source of heat generated during the process. The governing equations presented in the paper are not the correct form of the RANS equations. The authors are suggested to advise the relevant sources. If CFD software was used for the simulations, I suggest mentioning only the name and version of the software as well as the CFD models used along with references. The readers can be directed to the proper references as well as the guideline of the software.
12. All equations need references.
13. The text body presented at the beginning of section 4 is not relevant to simulation and boundary conditions. It is more of explaining the valve. Section 4.1 is also irrelevant to the simulation model. These can be merged with and presented as a separate sub-section of section 2 and named as “the problem definition”.
14. Explain the “sliding grid technique”.
15. Is it a 2D or 3D model?
16. Is the refrigerant R410A modelled in the liquid or gas state? What are the thermophysical properties of R410A?
17. Is it single-phase or tow-phase? In E-S tubes, the fluid is ideally in a vapour state (in reality it is a mixture of gas and liquid). Please mention any assumptions and simplifications made and reference them.
18. What was the software used for the numerical simulation? Please mention the version of each piece of software used.
19. What is the rationality behind using the mentioned values as boundary conditions? Are they based on a real AC system or taken from a reference? Please mention it.
20. What is the dynamic function used for boundary conditions? Please explain more about its maths and method of implementation in the simulation.
21. The mesh sensitivity analysis is required. Please provide the required information.
22. The governing equations are transient. Please also mention the time step and the sensitivity analysis for the time step size.
23. Please mention whether the “start position” is in cooling mode.
24. Please mention whether the pressure is absolute or gauge.
25. Please label E, S, C, and B tubes on all figures.
26. Some parts of the discussion section need further elaboration. For instance, why does pressure drop develops in a 180-degree turn? Please mention and explain the high- and low-pressure regions developed in the S tube in Figure 6.
27. Regarding Figure 6, It was mentioned that “There is no significant difference in the pressure on the surface of the slider”. Is it the internal of the external face? The face exposed to the D tube and chamber experiences a large pressure difference.
28. The main shortage of the paper is in the discussion section, where only graphs are presented and explained. There is a clear lack of discussion, reasons for observations, and comparison with previous studies.
29. In section 5.1, the graphs are explained but no results were drawn. Which design has the better performance? What are the pros and cons of each valve design?
30. Section 5.2 is not related to results and discussion and it must move before the results and discussion section. The experimental setup needs more explanation.
31. What is the uncertainty of the experiment?
32. The comparison with the experimental results better to be presented at the beginning of section 5 since it also shows the performance of the numerical model.
33. Please present Figure 11 as line graphs (similar to Figure 12). The differences can be presented with error bars on the graphs or in a separate table.
34. Please mention if the numerical or experimental pressure values are used in section 5.3.
35. What is the significance of figures 12 and 13? What results can be drawn?
36. The conclusion section must be supported by numerical values.
The English of the paper must be checked and improved. Poor English and a lack of technical academic language prevail in the paper. Some parts of the paper are unclear due to poor English.
Round 2
Reviewer 2 Report
There is no rebuttal submitted with the paper which makes navigating my comment is the paper to be difficult. Please submit a rebuttal.
I cannot see the response to some of my comments including the following
1. What is the novelty of this study? This must be addressed in the introduction section.
2. Explain the “sliding grid technique”.
3. What is the dynamic function used for boundary conditions? Please explain more about its maths and method of implementation in the simulation.
4. Please label E, S, C, and B tubes on all figures.
The quality of English language needs to be improved.
